# Validation of Inertial Measurement Units for Analyzing Golf Swing Rotational Biomechanics

**DOI:** 10.3390/s23208433

**Published:** 2023-10-13

**Authors:** Sung Eun Kim, Jayme Carolynn Burket Koltsov, Alexander Wilder Richards, Joanne Zhou, Kornel Schadl, Amy L. Ladd, Jessica Rose

**Affiliations:** 1Department of Orthopaedic Surgery, Stanford University, Stanford, CA 94305, USA; cjsekim@stanford.edu (S.E.K.); jcbk@stanford.edu (J.C.B.K.); zhoujy@stanford.edu (J.Z.); kornels@stanford.edu (K.S.); alad@stanford.edu (A.L.L.); 2Motion & Gait Analysis Lab, Lucile Packard Children’s Hospital, Palo Alto, CA 94304, USA

**Keywords:** golf, golf swing technique, biomechanics, rotational parameters, wearable sensors, inertial measurement units, golf coaching, performance analysis, real-time feedback

## Abstract

Training devices to enhance golf swing technique are increasingly in demand. Golf swing biomechanics are typically assessed in a laboratory setting and not readily accessible. Inertial measurement units (IMUs) offer improved access as they are wearable, cost-effective, and user-friendly. This study investigates the accuracy of IMU-based golf swing kinematics of upper torso and pelvic rotation compared to lab-based 3D motion capture. Thirty-six male and female professional and amateur golfers participated in the study, nine in each sub-group. Golf swing rotational kinematics, including upper torso and pelvic rotation, pelvic rotational velocity, S-factor (shoulder obliquity), O-factor (pelvic obliquity), and X-factor were compared. Strong positive correlations between IMU and 3D motion capture were found for all parameters; Intraclass Correlations ranged from 0.91 (95% confidence interval [CI]: 0.89, 0.93) for O-factor to 1.00 (95% CI: 1.00, 1.00) for upper torso rotation; Pearson coefficients ranged from 0.92 (95% CI: 0.92, 0.93) for O-factor to 1.00 (95% CI: 1.00, 1.00) for upper torso rotation (*p* < 0.001 for all). Bland–Altman analysis demonstrated good agreement between the two methods; absolute mean differences ranged from 0.61 to 1.67 degrees. Results suggest that IMUs provide a practical and viable alternative for golf swing analysis, offering golfers accessible and wearable biomechanical feedback to enhance performance. Furthermore, integrating IMUs into golf coaching can advance swing analysis and personalized training protocols. In conclusion, IMUs show significant promise as cost-effective and practical devices for golf swing analysis, benefiting golfers across all skill levels and providing benchmarks for training.

## 1. Introduction

The popularity of golf has grown internationally, with the number of participants reaching approximately 119 million in the US alone in 2023 [1]. Simultaneously, there is an increased demand for devices that can assist recreational golfers’ training to improve their golf swing techniques [2].

Implementing information gained from quantifying golf swing biomechanics analysis may assist in enhancing golf performance and reducing the risk of injuries [3]. The rotation of the upper torso and pelvis plays a crucial role in generating clubhead speed, which is a key factor in achieving greater distance and accuracy in golf shots. Biomechanical research has revealed that the X-Factor, or rotation between the upper torso and pelvic angles, during the swing, is closely linked to clubhead speed and power [4,5]. Additionally, the angles of the upper torso and pelvic rotation, along with their rotational velocities, influence swing efficiency and accuracy [6,7].

Biomechanical research on the golf swing has focused on the upper torso and pelvic rotation due to their significant associations with clubhead speed [8,9,10,11,12] and important differences between professional and recreational golfers [11,13,14]. Upper torso obliquity during the early downswing is also associated with clubhead speed and the direction of the ball outcome–greater obliquity is associated with faster clubhead speed [8] and less slice (curve to the right side) of the ball direction [15]. However, accurate calculation of these metrics typically necessitates a laboratory setting, which limits feasibility outside of the research setting [5]. As a result, there is a rising interest in developing portable and non-intrusive motion capture systems specifically designed for golf swing analysis [16]. Consequently, a training device capable of measuring upper torso and pelvic rotation could offer substantial performance improvements.

Inertial measurement units (IMUs) are increasingly being employed in the development of sports training devices, due to their wearable and low-cost features and their ease of use in the field [17]. A prior study successfully validated the angle of upper torso rotation measured by IMUs against the gold standard of 3D motion capture across four sports, including golf swing [18]. The study found that the IMUs measured the upper torso rotation angle with a deviation of less than 5 degrees compared to 3D motion capture, across all four sports. However, the upper torso rotation angle at the top of the backswing was found to be approximately 30 degrees, a stark contrast to the 100 degrees reported in prior studies for both professional and recreational golfers [11,13,14]. This discrepancy suggests the participants in the former study may not have been experienced golf players. Validating the use of IMUs across a broad spectrum of potential golfers, both male and female professional and recreational golfers, can provide a more robust test for data accuracy. Additionally, not only the angle of upper torso rotation but also other performance-related variables, such as pelvic rotation angle [19], X-factor [13], pelvic rotational velocity [20], and shoulder and pelvic obliquities [11,19], can provide a more comprehensive assessment of the utility of IMUs for measuring golf swing performance.

The purpose of this research is to investigate the accuracy of IMUs in measuring golf swing kinematics. Specifically, the aim was to determine the validity of IMU-based measures of the upper torso and pelvic rotation, pelvic rotational velocity, shoulder and pelvic obliquities, and X-factor during the golf swing. We hypothesized that IMU measures of golf swing rotation would correlate highly with the gold-standard 3D motion capture.

## 2. Materials and Methods

### 2.1. Participants

Testing was conducted at Stanford Medicine, Department of Orthopaedic Rehabilitation, Motion Analysis Laboratory (Redwood City, CA, USA). Based on an a priori power calculation, a total of 36 subjects would provide over 80% power to calculate the intraclass correlation coefficient for agreement between IMU and motion capture data to within a confidence interval width of 0.1, assuming ICCs of 0.8 or greater. To ensure a balanced cohort for validation, an equal number of amateur and professional golfers (18 each) and male/female gender (18 each) were recruited. The study was approved by the Institutional Review Board (eProtocol number: 63133), Stanford University, and consent was obtained from all volunteer participants.

### 2.2. Data Acquisition

Kinematic data were collected using a ten-camera optometric system with Cortex 9 Motion Capture Software for 3D motion analysis (Motion Analysis Corporation, Santa Rosa, CA, USA). The custom IMUs were recorded with in-house developed software written in Rust 1.71.0 and Python 3.10.0 at a sampling rate of 100 Hz. The average 3D residual error for the motion capture system was 1.2 ± 0.6 mm, indicating the degree of precision with which the system could reconstruct the location of each marker within the capture volume.

IMU data were captured using 2 IMU devices, with a sampling rate of 100 Hz. The IMUs used in the research were configured with a gyroscope setting full-scale range of 2000 degrees per second and an accelerometer full-scale range of 16 g, allowing for accurate recording of a wide range of angular velocities and accelerations during the golf swings. These IMUs were wirelessly connected to a lab computer via Bluetooth Low Energy, enabling real-time transmission of the collected data. The data were then stored for subsequent in-depth analyses.

### 2.3. Protocol

Four reflective markers (14 mm) were placed on the anterior superior iliac spines (ASIS) and acromions, bilaterally. This marker placement is a customized model in accordance with the swing variables validated in this study. The analysis of the upper torso and pelvis angular kinematics during the golf swing has used this marker placement in several prior studies [11,21,22]. A reflective tape (a radius of 15 mm) was also placed 5 cm proximal to the clubhead of the participants’ seven-iron golf club. A plastic practice ball, comparable in size to a standard golf ball, with the reflective tape attached, was utilized and placed on a synthetic grass mat. In addition, two custom IMUs were placed on the participant’s T1 and L4 vertebrae.

Participants were positioned in the laboratory so that their starting position had the upper torso and pelvis aligned with the frontal plane of the 3D motion capture space coordinate system. Each participant was given the opportunity to warm up with ‘easy’ golf swings and then was instructed to perform five ‘hard’ golf swings, from which a minimum of 3 representative swings were recorded.

### 2.4. Data Processing

For each golfer, a minimum of two to a maximum of three swings with no marker dropout were processed. Marker data were filtered using a Butterworth filter with a cutoff frequency of 12 Hz. Marker data from the ball and clubhead were not smoothed because the only interest was the time of swing phases. In addition, at impact, very large displacement occurs in those markers and smoothing would introduce an undesirable delay and reduction in magnitudes. Of note, to address this issue, some prior studies have used different cutoff frequencies for each marker, at which the clubhead was smoothed at 30 Hz [22,23].

Swing phases were delineated based on the clubhead and ball kinematics. The initiation of the backswing was identified when the clubhead’s vertical velocity exceeded 0.2 m/s. The initiation of the downswing was defined by the reversal of the clubhead’s vertical direction at the top of the backswing. Impact was defined as the time point immediately preceding the initial increase in ball velocity.

The rotation of the upper torso and pelvis were calculated as the rotation of the segments connecting the bilateral acromion and ASIS markers, respectively, along the vertical axis. The X-factor was calculated as the difference between the upper torso and pelvic rotation. The rotational velocity of the pelvis was calculated as the rate of change of the pelvic rotation [24,25,26,27]. Obliquity of the upper torso (S-factor) and pelvis (O-factor) was calculated as the angle between the horizontal plane and the segment connecting the acromion for the upper torso and the ASIS for the pelvis [11].

Each swing was normalized to a percentage of the golf cycle from the beginning of the backswing (0%) to the ball impact (100%). The end of follow-through (130%) was defined by the local minimum of vertical clubhead displacement following the circumduction of the club around the body during the follow-through phase. The swings of amateur golfers were plotted over the average curve of professional swings for comparative purposes.

IMUs placed on the T1 and L4 spinous processes were used to capture the raw accelerometer and gyroscope measurements of the golf swings. Orientation angles relative to the participant’s orientation at address were calculated using the Madgwick filter [28], resulting in angular measurements in the pitch, roll, and heading directions, where upper torso and pelvic rotations are represented by roll and the S-factor and O-factor are represented by heading. X-factor, or the relative rotation between the upper torso and pelvis, was determined by calculating the difference between the filtered and smoothed upper torso and pelvic angles. Finally, pelvic rotational velocity was calculated by calculating the rate of change in the filtered and smoothed roll angles of the pelvis.

### 2.5. Statistical Analysis

Statistical analyses were performed using a custom script on Python 3.10.0 (Python Software Foundation, Wilmington, DE, USA) for outlier detection and Bland–Altman analysis and SAS version 9.4 (Cary, NC, USA) for Intraclass Correlation and Pearson correlation.

#### 2.5.1. Outlier Detection

An outlier detection analysis was conducted to ensure the reliability and validity of subsequent analyses. Due to the time-series nature of the data, where each swing trial was composed of multiple frames, an outlier detection approach was employed, considering the entire swing [29]. The Z-scores for each biomechanical variable within each swing were calculated. Swings with frames that exhibited Z-scores above 3 or below −3 for any variable were identified and flagged as outliers and were excluded from further analysis, as previously recommended [29].

#### 2.5.2. Intra Class Correlations and Pearson Correlation

Rotational biomechanics of the golf swing was examined by calculating the Intraclass Correlation (ICC) and Pearson correlation coefficients between the 3D motion capture and IMU kinematic data. Six parameters, including upper torso and pelvic rotations, pelvic rotational velocity, S-factor, O-factor, and X-factor, were compared.

#### 2.5.3. Bland–Altman Analysis

Bland–Altman analysis was employed to further assess the agreement between 3D motion capture and IMUs. This statistical method involves plotting the mean of the two measurements against their difference, providing visual and numerical measures of agreement, including the mean difference and limits of agreement (LoA, defined as the mean difference ± 1.96 standard deviations of the difference) [30]. If the IMU and 3D motion capture are in good agreement, then the mean difference on the Bland–Altman plots should be near zero, indicating that IMU neither consistently over- or underestimates relative to 3D (i.e., low systematic bias). If the width of the LoA is narrow relative to the range of the data, this indicates relatively low random error between the two measurements. Other features of the Bland–Altman plots to look for are that the errors between the two measures and the variability in these errors are relatively consistent across the plot, indicating that these do not increase or decrease as the measurements get larger.

## 3. Results

### 3.1. Demographics

The study included a cohort of thirty-six golfers, comprising 18 professionals and 18 amateurs. The participants were evenly distributed between males and females. The mean age was 33.8 ± 15 years, with professionals being younger at 27.7 ± 10.4 years, and amateurs older at 39.9 ± 16.6 years. The mean height and weight were 173.8 ± 10.2 cm, and 71.6 ± 14.5 kg, respectively, with no substantial differences between professionals. The median handicap and interquartile range (IQR) across all participants was 4.5 [IQR: 0.8, 13.3]. The median handicap was significantly lower for professionals (0.5 [0.0, 2.8]) compared to amateur players (13.5 [IQR: 9.3, 18.8]). These demographic data provide an overview of the diverse study population, which comprises individuals with varying skill levels and physical attributes (Table 1).

### 3.2. Swing Results

A total of 108 swings were recorded from which a subset of five swings (4.6% of all trials) were identified as outliers based on z-scores above 3 or below −3. These outlier swings exhibited deviations primarily attributed to measurement errors and were excluded from further analysis. A total of 103 swings qualified and were included in the analysis.

To examine swing variation within a subject, a minimum of two swings (5 participants) or three swings (31 participants) were analyzed.

Figure 1 presents plots of the golf swing biomechanical variables derived from both IMU and 3D motion capture for each parameter. The plot illustrates the mean values over the golf swing cycle, starting with the beginning of the backswing, downswing, impact at 100%, and follow-through. Bands represent one standard deviation to provide a comprehensive view of the average and variability within the golf swing data.

#### 3.2.1. Intra Class Correlations and Pearson Correlation

The Intraclass Correlation Coefficients (ICCs) provided evidence of a strong correlation between the measurements collected by the IMU and 3D motion capture methods for the upper torso and pelvic rotation, pelvic rotational velocity, S-factor, O-factor, and X-factor (Table 2). Upper torso and pelvic rotation as well as S-factor exhibited a tighter, while pelvic rotational velocity, O-factor, and X-factor demonstrated wider variability (Figure 2). Correlation analysis of the second swings of each participant resulted in similarly high ICC values (pelvic rotation: 0.99, pelvic rotational velocity: 0.98, upper torso rotation: 0.99, S-factor: 0.99, O-factor: 0.91, X-factor: 0.94). Strong association is further supported by the Pearson coefficient and corresponding R^2^ values.

#### 3.2.2. Bland–Altman Analysis

The comparison of measurements between the IMUs and 3D motion capture was also assessed using Bland–Altman plots. The mean differences for all measures were close to 0, demonstrating that IMU neither consistently over- or underestimated the 3D measures (Table 3, Figure 3). The LoA was narrowest for torso and pelvic rotation, only around 8–10% relative to the range of these measures. The width of the LoA was greatest for O-factor but was still only a third that of the range of the data for this measure. None of the Bland–Altman plots showed evidence that the error in the data or the variability in the error varied appreciably over the range of measurement.

## 4. Discussion

The present study aimed to investigate the accuracy of inertial measurement units (IMUs) in measuring the rotational biomechanics of the golf swing. We compared IMU-based measures with the gold standard of 3D motion capture, seeking to determine the validity of IMUs for assessing upper torso and pelvic rotation, pelvic rotational velocity, shoulder and pelvic obliquities, and X-factor during the golf swing.

Our findings indicate that IMUs provide a highly accurate alternative for capturing golf swing kinematics, exhibiting strong correlations with lab-based 3D motion capture systems as assessed by ICC and Pearson correlation analysis. Mean differences and limits of agreement (LoA) across various parameters revealed the overall agreement between the two methods. It is worth noting, however, that variations existed across different aspects of the swing.

The mean differences between IMU and 3D motion capture data were all close to 0. Relative to the range of the data, the mean difference was smallest for pelvic rotation (0.2%) and largest for O-factor (2.8%). The LoA were narrowest relative to the range of the data for the rotational measures (8.1% for upper torso rotation and 10.2% for pelvic rotation), and widest for X-factor (25.0%) and O-factor (33.4%). Given the range of rotation and rotational velocity, the absolute mean differences detected are negligible, indicating high accuracy. This underscores the importance of precision in capturing golf swing rotational biomechanics, while also affirming the robustness of the method, despite minor deviations.

Post-processing and filtering of IMU measurements exert a significant impact on IMU-derived metrics. This effect is particularly pronounced for derived values such as pelvic rotational velocity, which is computed by differentiating the pelvic rotational angles over time. Figure 4 illustrates this phenomenon by depicting a single trial that demonstrated notable inconsistency at the beginning of the swing, most likely attributed to a sudden movement shifting the IMU at the beginning of backswing due to loose attachment. The rest of the measurement for the swing, however, appears to be consistent and accurate. This discrepancy negatively impacts the agreement between the two methods and may lead to disparities in the data analysis. A refined filtering method could enhance the precision of the measurements, including those of the pelvic rotational velocity. Consequently, this could further improve the congruence between the IMU and 3D motion capture data.

Precise placement of IMUs is vital for the accurate measurement of upper torso and pelvic rotation. The disparity in locations for calculating upper torso rotation, such as the T1 vertebra for IMU versus the segment between the two acromion markers for 3D marker-based analysis, may account for slight variations in the measurement of upper torso rotation between the two methods. These variations can also be influenced by specific movement characteristics, such as upper torso protraction during the follow-through phase of a swing, where the spine may stop rotating but the upper torso continues.

Our findings have significant implications for golf swing analysis, particularly in the context of training and performance improvement. Traditional 3D motion capture for golf swing analysis has been mainly limited to laboratory settings, making it challenging for recreational golfers to access biomechanical insights and performance feedback outside of research settings. Wearable IMU devices offer portable and cost-effective access, enabling golfers to receive immediate feedback and guidance on their swing techniques both on the driving range and the golf course.

The integration of IMUs into existing golf coaching and training technologies presents an exciting opportunity [5]. IMUs offer comprehensive and instantaneous feedback, which can complement traditional coaching methods and accelerate advancements in golf performance. Furthermore, researchers and practitioners can gain valuable insights into the biomechanics of each individual golf swing, which, in turn, can lead to the development of new injury prevention strategies for golfers.

A promising avenue these findings uncover is the possibility of creating a single score swing performance index, which could be computed solely utilizing IMUs. Such an index would gauge the disparity between an individual’s swing and the pro benchmark swing, enabling golfers to assess their performance, pinpoint areas for enhancement, and monitor their progress over time [7].

As golf continues to grow in popularity, the sheer number of participants and the portable nature of wearable IMUs present an opportunity for collecting large-scale golf swing data [31]. Analyzing big data can reveal valuable insights and patterns in swing mechanics across different skill levels and playing styles [32]. In addition, the application of artificial intelligence (AI) in analyzing extensive golf swing data offers great promise in interpreting large volumes of swing-related information, thereby enabling highly individualized and nuanced suggestions to improve golf swings. By identifying key performance factors and swing patterns, golfers and coaches can tailor training programs for individual needs and optimize performance strategies.

Potential future directions involve the formulation of personalized training protocols utilizing IMU-based metrics. Through the comparison of an individual golfer’s swing data with the established benchmark patterns observed in professional golfers, a customized training regimen can be devised to target the unique requirements of each golfer.

While our findings hold significant implications for golf swing analysis and performance improvement, it is essential to address certain limitations in our study. The observed small variations in agreement between IMUs and 3D motion capture for specific parameters may be influenced by individual swing styles, IMU placement accuracy, and environmental factors, such as radio interference and data loss during transmission. To overcome these limitations, future research should focus on standardizing IMU placement and calibration procedures and investigating the impact of environmental factors. By addressing these challenges, the accuracy and reliability of IMU-based measurements can be further improved, unlocking the full potential of wearable devices for golf swing analysis and offering practical insights for golfers across skill levels.

In conclusion, our research revealed that IMUs can provide a reliable and accurate method for measuring rotational biomechanics in the golf swing. The accuracy and validity of IMUs when compared to the gold-standard 3D motion capture, across a wide spectrum of golfers, highlight their potential use as portable, non-invasive, and cost-effective devices in golf swing analysis and training. This supports the potential for IMUs to facilitate biomechanics analysis outside of lab-based settings, thereby enabling real-world applications, such as on-course training and performance enhancement for both professional and recreational golfers. These findings pave the way for future research to further refine and expand the use of IMUs in sports biomechanics.

## Figures and Tables

**Figure 1 sensors-23-08433-f001:**
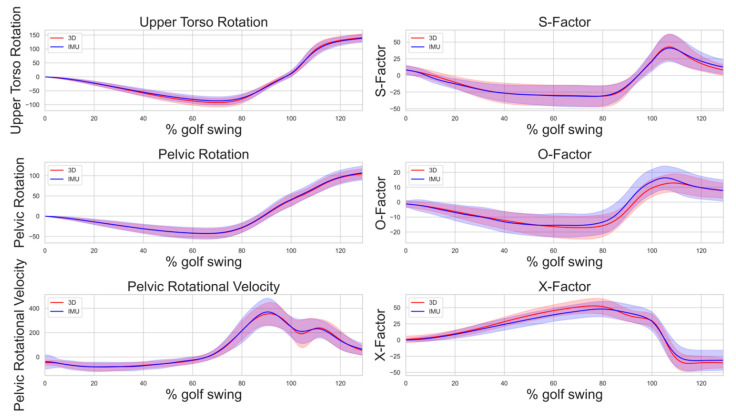
Comparison of 3D motion capture and IMU measurements across different swings; shaded regions represent 95% confidence intervals (CIs) around the plotted lines for the golf swing cycle, with impact at 100%.

**Figure 2 sensors-23-08433-f002:**
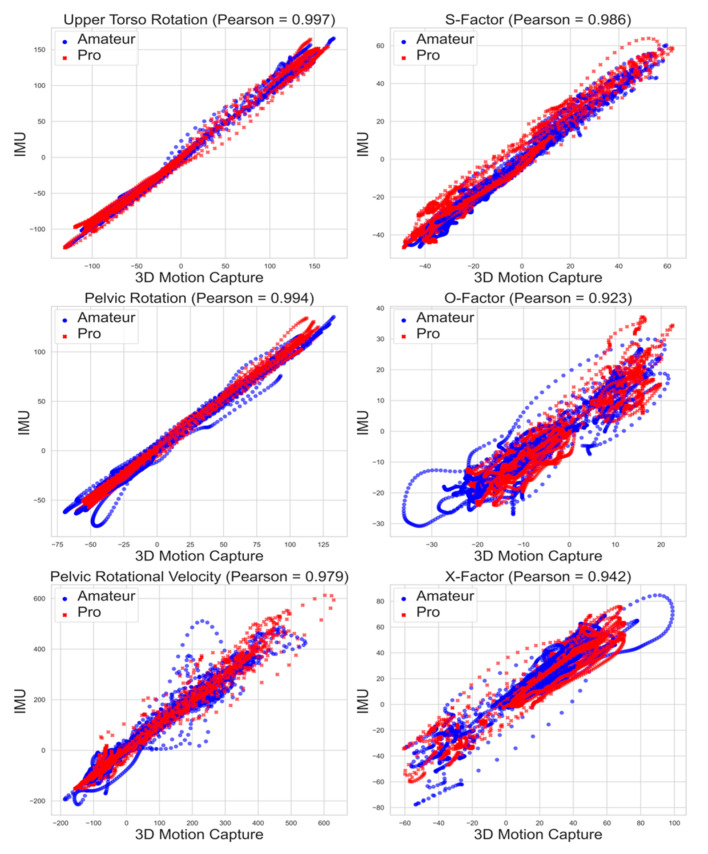
Scatterplots showing the relationships between IMU and 3D motion capture measurements.

**Figure 3 sensors-23-08433-f003:**
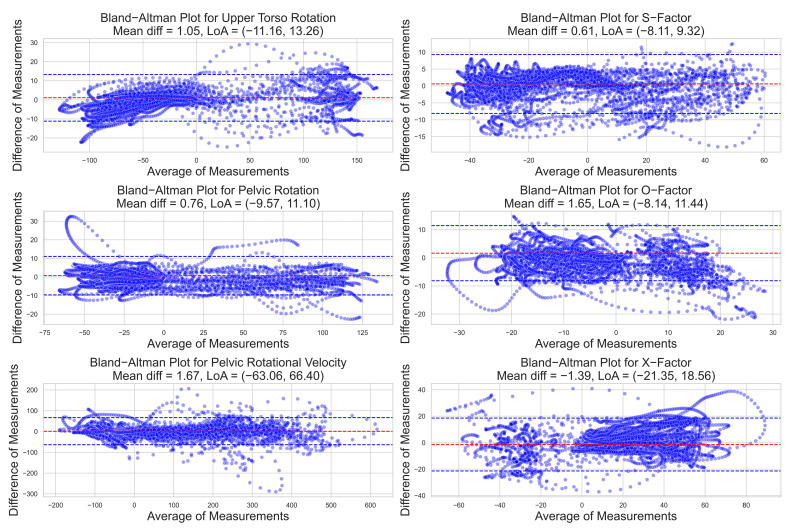
Bland–Altman plots comparing the measurements of golf swing parameters obtained from IMUs and 3D motion capture. Mean difference (red line) and limits of agreement (blue lines) are displayed, showing the agreement between the two measurement methods. The plots help identify systematic bias and the degree of random error between the methods.

**Figure 4 sensors-23-08433-f004:**
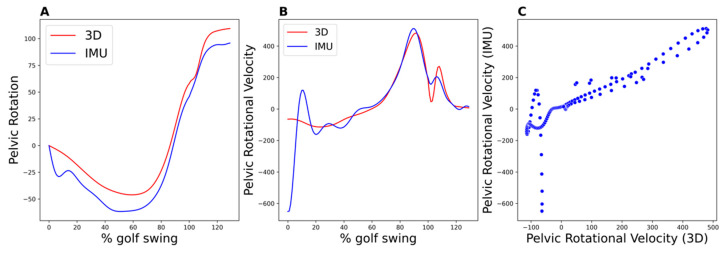
Artifact observed on IMU-derived metrics for a single trial. (**A**): Pelvic rotation, (**B**): pelvic rotational velocity as a function of time for both IMU and 3D motion capture measurements. (**C**): Scatter plot comparing pelvic rotational velocity derived from 3D motion capture (*x*-axis) against the IMU-derived pelvic rotational velocity (*y*-axis).

**Table 1 sensors-23-08433-t001:** Descriptive Statistics of Demographic and Physical Characteristics for Professional and Amateur Golfers. SD = standard deviation.

	All Mean (SD)	Pro Mean (SD)	Amateur Mean (SD)
**Age**	33.8 (15.0)	27.7 (10.4)	39.9 (16.6)
**Height (cm)**	173.8 (10.2)	174.4 (10.4)	173.1 (10.2)
**Weight (kg)**	71.6 (14.5)	70.5 (14.6)	72.7 (14.7)

**Table 2 sensors-23-08433-t002:** Pearson correlation coefficients and Intra Class Correlation (ICC) comparing IMU and 3D motion capture methods. All Intraclass and Pearson correlations were highly significant (*p* < 0.001).

	ICC (95% CI)	Pearson (95% CI)	R-Square
**Upper Torso Rotation**	1.00 (1.00, 1.00)	1.00 (1.00, 1.00)	0.99
**Pelvic Rotation**	0.99 (0.99, 0.99)	0.99 (0.99, 0.99)	0.99
**Pelvic Rotational Velocity**	0.98 (0.98, 0.98)	0.98 (0.98, 0.98)	0.96
**S-factor**	0.99 (0.98, 0.99)	0.99 (0.98, 0.99)	0.97
**O-factor**	0.91 (0.89, 0.93)	0.92 (0.92, 0.93)	0.85
**X-factor**	0.94 (0.93, 0.94)	0.94 (0.94, 0.95)	0.89

**Table 3 sensors-23-08433-t003:** Bland–Altman analysis comparing 3D motion capture and IMU measurements for each parameter. The mean difference and limits of agreement are provided for each parameter, quantifying the agreement between the two measurements.

	Mean Difference	Mean Difference % of Range	LoA (Lower)	LoA (Upper)	LOA % of Range
**Upper Torso Rotation (deg)**	1.05	0.3%	−11.16	13.26	8.1%
**Pelvic Rotation (deg)**	0.76	0.4%	−9.57	11.10	10.2%
**Pelvic Rotational Velocity (deg/s)**	1.67	0.2%	−63.06	66.40	15.9%
**S-factor (deg)**	0.61	0.5%	−8.11	9.32	15.7%
**O-factor (deg)**	1.65	2.8%	−8.14	11.44	33.4%
**X-factor (deg)**	−1.39	−0.9%	−21.35	18.56	25.0%

## Data Availability

The datasets generated and/or analysed during this ongoing study are not currently publicly available per the experiment protocol approved by the Institutional Review Board.

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
