# Peer review of "Validation of Inertial Measurement Units for Analyzing Golf Swing Rotational Biomechanics"

_sensors, 2023, doi:10.3390/s23208433_

Round 1
Reviewer 1 Report
The figures are suggested to be polished since the text size is small and difficult to read.
No comments.
Reviewer 2 Report
The authors should be commended for producing a very timely paper much appreciated by golfers. Whilst the paper is interesting it should be noted a few lab based unpublished feasibility studies do exist which potentially may deem this paper not original. However, this is a well presented paper that couldhave benefited from a pictorial illustration of the overall set up including a representative picture of a participant. The strength of this paper is the fact that it is succinctly presented showing highly useful data that golfers can identify with and benefit from. Some minor concerns have arisen which the authors should be able to address.
The protocol is concisely explained. What 3D model did you use as the gold standard? Did you use the plug in gait model for example or did you develop your own model using a customised marker set with probably cluster markers too? This needs to be clearly stated in the manuscript so that the protocol could be repeated by other golf researchers.
How was the reflective marker attached to the plastic practice ball? Did the marker not fly off the ball on impact with the club? I am aware you may not have been interested in other performance parameters such as club head speed and ball speed but some clarity here would be much appreciated.
Line 226 - 'O' Factor. 'O' should be in caps.
In general, the data processing techniques seems fine and overall, the paper is well written.
Reviewer 3 Report
sensors-2615695
Reviewer comments
In the submitted paper, the author(s) examined the validity of an IMU-based method for the analysis of the angular kinematics of the golf swing. The validity of the method was examined using a 3D motion analysis system as reference. Male and female professional and amateur golfers were examined. Results revealed that no differences exist regarding the error values across individual angles within a single stance.
The manuscript is well written, but there are some concerns that need to be addressed.
General Comments
- The parameters concerning obliquity and their importance on the effectiveness of the golf swing should be introduced with more detail within the Introduction.
- Provide the rationale of selecting subgroups of both professional and amateur female and male golfers but treating their data as a single group. It is suggested to provide evidence that the golf swing rotational biomechanics is not different between professional and amateur golfers and between males and females.
- It is suggested to include section 3 (Statistical analysis) within section 2 (Materials and Methods).
Specific comments:
Abstract
- L18-19: Interclass Correlation coefficients ranged … Report also the 95% Confidence Intervals for the ICCs.
Introduction
- See the respective General Comment.
Materials and methods
- See the respective General Comment.
- Provide the softwares used for the data acquisition and analysis, as well for the statistical analyses.
- L91-92: Please mention the anthropometric model used as reference for the placement of the markers. Provide also the dimensions of the markers.
- L101: State the maximum number of trials analyzed for each participant.
- L103: Provide the rational for not smoothing these markers. Could the inclusion of their smoothed trajectories provide additional insight regarding the separate phases of the swing, its effectiveness and possible differences among the subgroups?
References
- Provide all references according to the journal’s guidelines [i.e., use of abbreviated journal titles].
- L340-341: Some reference details seem missing for ref. #14. The same at L362 (ref. #24).
- L368: delete “Amp;”.

Round 2
Reviewer 3 Report
sensors-2615695-R1
Reviewer comments
In the resubmitted version of the manuscript, the author(s) did an exceptional work to address adequately the concerns raised in the initial round of reviewing. There a couple of minor issues still need to be addressed.
Comments
- L141-142: State the version of the IBM SPSS and the Python software.
- L141-142: Add also the manufacturer details for SPSS (ΙΒΜ Corp., Armonk, NY, USA) and the Python Software Foundation.
- L141-142: It is suggested to add which analysis was conducted with the use of which software.
- L377: Biomech.
